# Evaluation of the shielding initiative in Wales (EVITE Immunity): protocol for a quasiexperimental study

Bridie Angela Evans [1,2] Ashley Akbari [1] Rowena Bailey,[1] Lesley Bethell,[1] Samantha Bufton,[3] Andrew Carson-Stevens [4] Lucy Dixon,[1] Adrian Edwards [4] Ann John [1] Stephen Jolles,[5] Mark Rhys Kingston,[1,2] Jane Lyons [1] Ronan Lyons [1] Alison Porter [1,2] Bernadette Sewell,[6] Catherine A Thornton,[1] Alan Watkins [1] Tony Whiffen [3] Helen Snooks[1,2]

[1]Swansea University Medical School, Swansea, UK
[2]PRIME Centre Wales, Swansea University Medical School, Swansea, UK
[3]Knowledge and Analytical Services, Welsh Government, Cardiff, UK
[4]Division of Population Medicine, Cardiff University, Cardiff, UK
[5]University Hospital of Wales, Cardiff, UK
[6]Swansea Centre for Health Economics, Swansea University, Swansea, UK

**Correspondence to**
Dr Bridie Angela Evans;
b.a.evans@swansea.ac.uk

## ABSTRACT

**Introduction** Shielding aimed to protect those predicted to be at highest risk from COVID-19 and was uniquely implemented in the UK during the COVID-19 pandemic. Clinically extremely vulnerable people identified through algorithms and screening of routine National Health Service (NHS) data were individually and strongly advised to stay at home and strictly self-isolate even from others in their household. This study will generate a logic model of the intervention and evaluate the effects and costs of shielding to inform policy development and delivery during future pandemics.

**Methods and analysis** This is a quasiexperimental study undertaken in Wales where records for people who were identified for shielding were already anonymously linked into integrated data systems for public health decision-making. We will: interview policy-makers to understand rationale for shielding advice to inform analysis and interpretation of results; use anonymised individual-level data to select people identified for shielding advice in March 2020 and a matched cohort, from routine electronic health data sources, to compare outcomes; survey a stratified random sample of each group about activities and quality of life at 12 months; use routine and newly collected blood data to assess immunity; interview people who were identified for shielding and their carers and NHS staff who delivered healthcare during shielding, to explore compliance and experiences; collect healthcare resource use data to calculate implementation costs and cost–consequences. Our team includes people who were shielding, who used their experience to help design and deliver this study.

**Ethics and dissemination** The study has received approval from the Newcastle North Tyneside 2 Research Ethics Committee (IRAS 295050). We will disseminate results directly to UK government policy-makers, publish in peer-reviewed journals, present at scientific and policy conferences and share accessible summaries of results online and through public and patient networks.

## INTRODUCTION

Shielding was introduced early during the COVID-19 pandemic across the UK. It was intended to protect those thought to be at

---

### STRENGTHS AND LIMITATIONS OF THIS STUDY

⇒ This research will provide the first population-scale national assessment about effects of shielding on COVID-19 infection rate, mortality, serious illness, use of National Health Service resources, health-related quality of life and behaviour.

⇒ In this study, we will develop a logic model for shielding, providing the first summary of the rationale for this internationally unique and untested public health intervention and underpinning interpretation and contextualisation of our study findings.

⇒ This study will use mixed methods to understand processes, effects and costs of shielding at national and individual level, including assessment of impact of immunological status on outcomes.

⇒ The primary limitation of the EVITE Immunity study is construction of our matched cohort; we will undertake validation checks to understand differences between groups and allow appropriate adjustments for these in our statistical analysis plan.

⇒ The development of the EVITE Immunity study has involved people with direct experience of shielding from the outset, with public contributors represented across all aspects of the study, reflecting strong views that evidence about effects of shielding is needed.

---

highest risk of serious harm should they catch COVID-19 because of pre-existing conditions such as cancer or treatment such as immuno-suppressive medications. It became apparent at an early stage of the pandemic that the virus was disproportionately affecting some parts of the general population, including older people[1] and patients with pre-existing conditions such as cardiovascular disease, respiratory disease and cancer.[2 3] A cohort study of over 17 million primary care records in England[4] confirmed the association between diagnoses such as diabetes and asthma and risk of death from COVID-19 and

also highlighted the risks associated with deprivation, old age and being male and black or South Asian.

In response to increasing transmission and deaths from COVID-19, governments across the UK nations developed methods to identify people thought to be most vulnerable to COVID-19 infection, hospital and intensive care unit (ICU) admissions, serious illness or death.[5–8] These people were selected for advice to shield, before and in addition to more general lockdown measures introduced across the population. In March and April 2020, Public Health England and Public Health Wales advised individuals by letter, text or phone call to strictly self-isolate, even from people within the same home, for a period of 12–16 weeks. Support such as food parcels, prescription delivery and priority supermarket shopping slots were provided and individuals were eligible for Statutory Sick Pay.[6 7] Shielding along with other lockdown restrictions, eased temporarily from late summer 2020 and was then reinstated shortly before Christmas 2020, with some variation by and within nation, until spring 2021.

Shielding aimed to protect those judged to be at highest risk of serious harm should they become infected with COVID-19.[5] The mechanism for avoiding harm was to avoid infection. Clinically extremely vulnerable (CEV) patients with diagnoses including cancer, serious heart conditions, respiratory problems and receiving certain treatments such as transplants and immunosuppressant medications were identified through algorithms and individual clinical screening methods from routine national and local National Health Service (NHS) data sources.[6–8] Following a refinement of the medical criteria for shielding in May 2020,[9] the shielding population increased. In England, it was estimated to be 2.2 million and in Wales 133 000 people at July 2020.[10 11] Shielding in the UK cost £308 million to deliver in the first 4 months.[12]

Shielding is a new intervention, uniquely used in the UK during the 2020 pandemic without prior evidence of effects on health outcomes or behaviour including intended and unintended consequences.[12–16] The WHO recognises that some people are at higher risk than others from COVID-19, but states that 'all must act to prevent community spread'.[17] It encouraged measures—including physical distancing, handwashing and stay-at-home advice—to limit transmission and protect populations to ensure that health services can sustain increased demand for patient care and treatment.

Evidence is now emerging of effects of shielding on: physical and mental health; well-being and quality of life including social isolation, loneliness and anxiety; access to medical care.[18–21] Higher COVID-19 rates among people who shielded are also reported.[22 23] There may be additional secondary effects on physical and mental health, across the shielded population or in subgroups such as the very elderly, people in different clinical condition groups and ethnic minorities. Questions

remain about whether the screening process, which involved complex stratification based on modelling to account for ethnicity, deprivation and comorbidities, was the most appropriate approach.[5]

There is a pressing need for rigorous population level evidence to build on early findings from the small-scale studies undertaken so far. It is well known that healthcare interventions do not always achieve intended effects.[24 25] High-quality evidence about effects of shielding advice on COVID-19 infection rate, mortality, serious illness, use of NHS resources, health-related quality of life and behaviour is therefore urgently required to inform policy and practice throughout this and any future pandemic.

We describe our protocol to evaluate the effects and costs of shielding in Wales where we will extend existing data linkage to COVID-19 diagnosis and antibody (serology) laboratory results, adopting a quasiexperimental matched cohort linked data study design to answer our research questions. As the shielding policy in Wales broadly replicated the policy in the rest of the UK, evidence from this evaluation will inform policy development and delivery in England as well as the devolved nations.

## Study aim
To measure effects and costs of shielding to protect members of the general population at highest risk of serious illness or death from COVID-19 in Wales.

## Objectives
1. Capture the rationale for UK shielding.
2. Assess effects of shielding in the general population and subgroups in terms of deaths, hospitalisations, safety and self-reported health.
3. Assess the infection levels and immunity within the shielded and control populations as a whole.
4. Explore behaviour, adherence and safety concerns relating to shielding.
5. Assess the costs of the shielding intervention against its consequences.
6. Understand the experiences and views of healthcare providers in relation to the shielding intervention and perceived effects, including healthcare associated harms.

## METHODS AND ANALYSIS
See table 1 for an overview of objectives, methods and outputs.

## Design
Quasiexperimental evaluation.

## Participant identification and participation (objectives 2–5)
Eligible participants will include those identified as CEV in Wales between March and May 2020 and who were advised to shield (shielded cohort). We will use individual-level population-scale anonymised

**Table 1** Summary of methods and outputs against study objectives

| Objective | Method | Output |
|---|---|---|
| 1. Capture rationale for UK shielding. | Interviews with policy-makers. | Logic model to describe components, outcomes and mechanisms of shielding. |
| 2. Assess effects of shielding in the general population and subgroups in terms of deaths, hospitalisations, safety and self-reported health. | Comparison of anonymised routine data between shielded population and comparator group. | Comparative outcomes for shielded and matched non-shielded people: COVID-19 tests, PCR-confirmed infections and deaths. All-cause mortality. Emergency department attendances, emergency hospital admissions, days spent in hospital, ICU admissions and days spent in ICU. |
| | Questionnaire to stratified random sample of intervention and control groups. | Self-reported health-related quality of life, anxiety, depression and loneliness. |
| 3. Assess the infection and immunity within the shielded and control populations as a whole. | Analysis of routine records of blood tests in shielded population. Analysis of blood samples from stratified random subsample of intervention and control groups. | Detailed analysis of COVID-19 antigen and antibody test results within the shielded and comparator populations, to understand: ▶ Effects of shielding on infection and immunity across the population and for clinical subgroups including cancer. ▶ Impact of immunological status on outcomes in the shielded and non-shielded population. |
| 4. Explore behaviour, adherence and safety concerns relating to shielding. | Interviews with 40 individuals and carers/household members who were shielded. | Experiences of shielded people and their carers or household members during the COVID-19 lockdown, including behaviour, emotional effects and safety concerns. |
| 5. Assess costs of the shielding intervention against its consequences. | Investigation of costs and cost–consequences of managing and delivering shielding. | Costs of intervention implementation and subsequent healthcare resource use compared with consequences and outcomes, net monetary benefit. |
| 6. Understand the experiences and views of healthcare providers in relation to the shielding intervention and perceived effects, including healthcare associated harms. | Interviews with 30 community-based health professionals. | Experiences of clinicians delivering care to shielded population during pandemic. |

ICU, intensive care unit.

data within the Secure Anonymised Information Linkage (SAIL) Databank to identify those shielded and their linked Electronic Health Records (EHR) from routinely collected NHS data sources, Office for National Statistics data and other health and administrative data, including the Wales Multimorbidity Cohort COVID-19 extension, under existing Information Governance Review Panel (IGRP) approvals.[26] We will also identify a cohort to match those identified for shielding with partners in Digital Health and Care Wales (DHCW), through cohort 'propensity' matching variables including age, sex and historic health service utilisation. We will create a stratified random sample from each of the two cohorts for questionnaire distribution and blood sample collection.

Prospectively collected data from questionnaires and bloods will be linked back into the SAIL Databank for anonymised linkage. Figure 1 describes data flow. Figure 2 describes recruitment and participation.

## Intervention

Individuals on the Shielded Persons List identified as CEV were sent advice (table 2), by letter (dated 24 March 2020) email or text, to stay at home for 12 weeks and 'do not go out at all' plus to minimise contact with anyone in the same household or visiting to provide care, 'even friends and family'.[27] Correspondence after the first 12-week shielding period reflected an easing, then reinstatement, then more easing.[28–31]

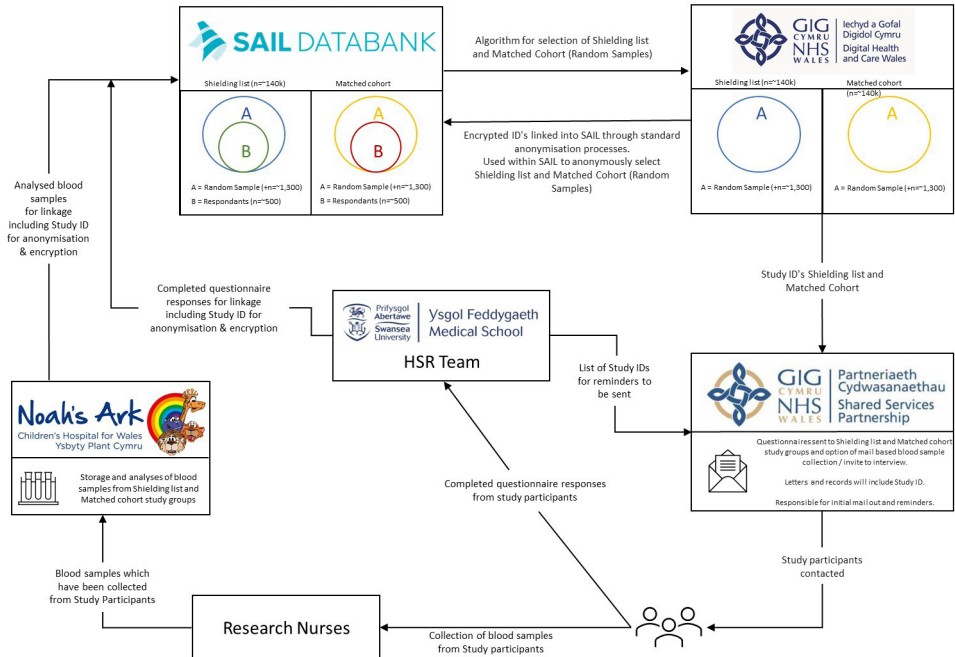

**Figure 1** Data flow visualisation.

## Methods for each objective

### Objective 1: rationale for UK shielding

We will conduct virtual interviews and focus groups with senior policy-makers and clinicians from Public Health and Chief Medical Officers' teams in Wales and England. We will encourage participants to consider the aim of the shielding policy, components of the intervention, the way it was intended to work and any perceived risks or unintended consequences of shielding. We will invite comments from participants on a draft logic model, including: components of the intervention; mechanisms for change (how the intervention was expected to work); expected outcomes and impacts, including harms. The draft logic model was prepared by the study team based on published information. We will record and transcribe interviews, with participants' consent, and will use data to refine and agree a logic model[32 33] to guide interpretation of study findings.

### Objective 2: effectiveness of shielding

We will create a matched electronic cohort and compare demographics and clinical characteristics to understand differences between the two groups; it will not be possible to achieve a perfect match, but characterisation of differences, incorporated into our statistical analysis plan, will allow appropriate adjustments when answering our research questions. We know that those warranting shielding will have higher utilisation rates. The rationale for matching on healthcare utilisation is to identify people who should have been shielding (but were missed due to initial selection of conditions based on prioritisation for influenza vaccination and/or administrative error) who can be matched on propensity, to create as similar as possible a comparator group (in the absence of randomisation) and use a difference in difference approach to

estimate effectiveness, comparing pre intervention and post intervention health service utilisation rates.

1. We will use anonymised individual-level linked routinely collected anonymised EHR data to compare outcomes between the two cohorts—COVID-19 infections, deaths, hospitalisations, immunity status, safety and costs up to 12 months. Inclusion of approximately 120 000 people in each cohort—from date of their addition to the shielding list between 23 March and 31 December 2020; and from 23 March 2020 for the matched control groups: with follow-up of outcomes up to 1 year—gives ample power to detect small differences (standardised statistical effects as small as 0.05, 90% power, 5% significance) in outcomes between groups and between most subgroups. For instance, 3%–5% of each cohort will be recorded as belonging to a black, Asian or other minority ethnic group, allowing comparison of outcomes between up to 6000 people per subcohort; larger numbers will be included in clinical subgroups such as cancer, heart disease, diabetes. We recognise it will not be possible to completely mirror the shielded group in our matched cohort. Our statistical analysis plan will incorporate ways of characterising differences and making appropriate adjustments.

2. We will examine self-reported outcomes at 12 months. We will distribute 1333 postal questionnaires (with online response option) (online supplemental appendix 1) to a stratified random sample in each of the shielded and non-shielded (matched) cohorts to achieve 533 responses in each. Questionnaires will include: the health-related quality-of-life measure (SF12);[34] measures of common mental disorders, anxiety and depression (PHQ9, GAD7);[35 36] safety concerns and

## EVITE study participant recruitment flowchart

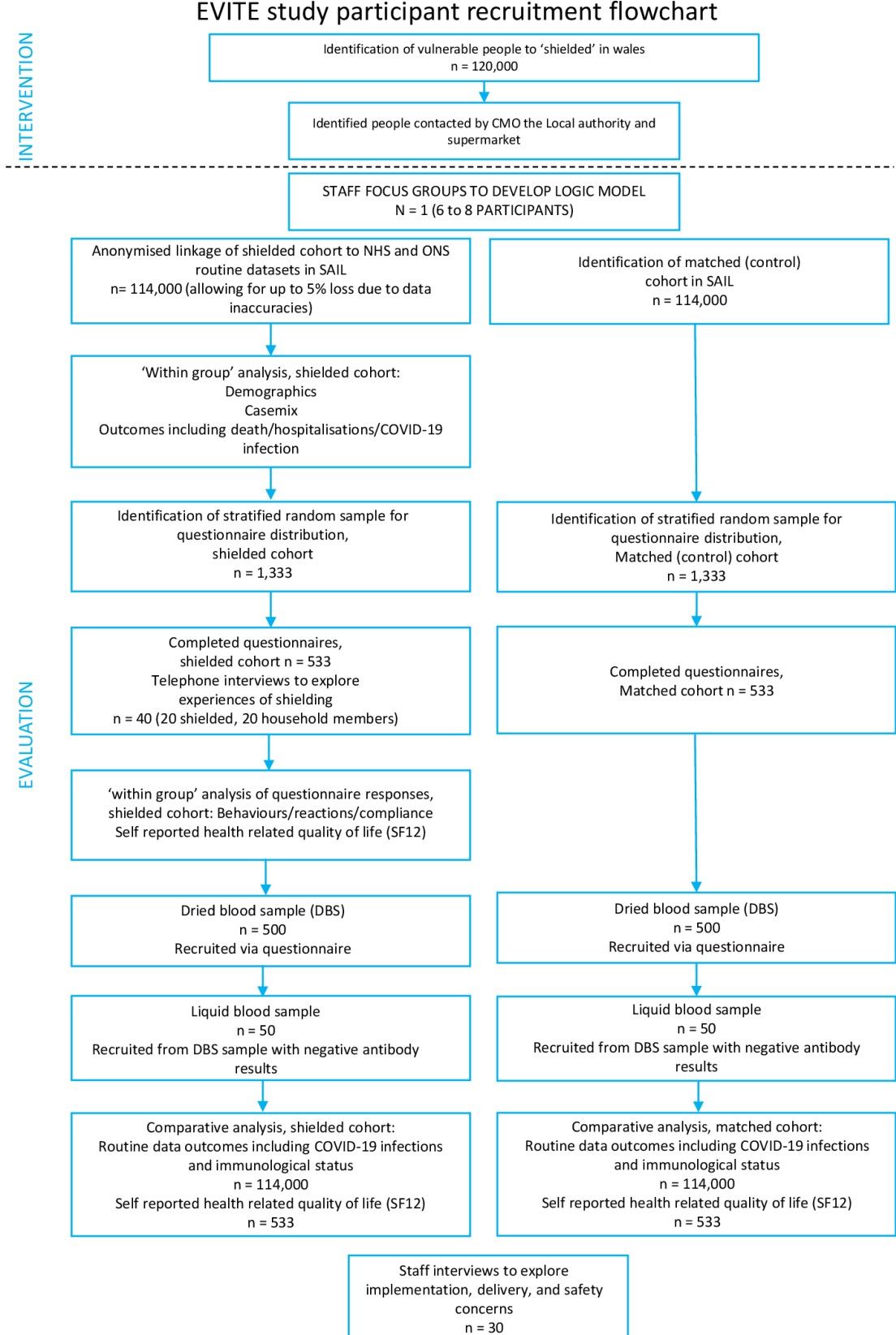

**Figure 2** Study participant recruitment flow chart.

behaviour during COVID-19 lockdown (staying home, isolating—including within home). Our sample size is sufficient to detect an average difference of 2.5 in SF12 component scores (standardised statistical effect 0.2, 90% power, 5% significance). Our Wales NHS partners, who hold contact details of individuals, will send

| Table 2 | Shielding advice given to clinically extremely vulnerable (CEV) people |
|---|---|
| Key points | Do not leave the house to go to work or to see other people |
| | Avoid being in the same room as another person |
| | Keep three steps away from another person in the home |
| | Avoid sharing kitchen, bathroom or bedroom facilities with others in the home |
| | Eat meals separately from other household members |
| | Be aware that hospital appointments and treatment may be postponed or cancelled |
| Support | Letters included details of available support, including how to obtain food, prescriptions and other information |
| Repeat advice | Letters were sent to CEV people throughout the pandemic which updated current advice. |
| | Correspondence after the first 12-week shielding period reflected an easing of some of the above points until January 2021 when shielding advice was reinstated. Advice remained, with some amendments, until August 2021. Shielding officially ended in March 2022. |

the questionnaires and assign each an anonymised ID number. This will ensure that the research team has no identifying information about participants at any stage of the evaluation.

## Objective 3: impact of immunological status on outcomes in the shielding population

We will analyse data from tests already undertaken using anonymised population-scale linked EHR data sources including COVID-19 PCR data held in SAIL to make a broad assessment of immunity, immunological status, infections and antibiotic use. Immunological data will include:

▶ Full blood count (FBC)—haemoglobin, platelets, white cell count, neutrophils, lymphocytes.
▶ Liver function test (LFT)—calculated globulin (total protein—albumin) Laboratory Information Management System (LIMS) test code B3062 (analysis by low and high calculated globulin).
▶ C reactive protein RP LIMS code B3023.
▶ Procalcitonin.
▶ Immunoglobulins (Ig)—IgG, IgA and IgM LIMS test code B3054.
▶ Serum electrophoresis.
▶ Glycated Haemoglobin.
▶ Renal function B5373.

We will also assess availability of less frequently tested immunological data:

▶ Lymphocyte subsets—cluster of differentiation (CD): CD3, CD4, CD8, CD19 and CD56.

▶ Specific antibodies—haemophilus influenzae B, tetanus, pneumococcus and SARS-CoV-2.

We will invite people approached by postal questionnaire (2.2) to provide blood samples for linkage and extended analysis. We will collect dried blood spots from those who indicate consent on completed questionnaires (≥500 in each study arm). This self-administered test will be delivered and returned by post. We will identify a subgroup (50 in each study arm) and invite them to provide a liquid blood sample to investigate T-cell immunological responses. The liquid blood sample will be collected by a qualified health professional in the individual's home.

## Objective 4: behaviour, adherence and safety concerns of people who shielded

We will interview 40 people on the Shielded Persons List, their carers and/or household members, from those who agree to be contacted after completing the questionnaire (2.2) and then consent to take part. Interviews will be by telephone or online (eg, Zoom), recorded and transcribed with their consent. We will explore individual experiences during 2020, including behaviour, physical and mental health and also safety concerns (ie, an event or situation where something went wrong or not as expected while receiving or trying to receive healthcare).[37 38] People with experience of shielding on our study team will codevelop the interview questions (online supplemental appendix 2). The study team will ensure interviewees reflect the range of people included in the shielding intervention in age, sex, health status, ethnic group and place of residence in Wales. We recognise that our sample may include individuals who have been bereaved and we will signpost to appropriate support.

## Objective 5: costs of the shielding intervention against its consequences

We will investigate implementation costs, including costs incurred in identifying those asked to shield and in managing shielding processes. We anticipate this will include costs of:

▶ Developing and implementing algorithms (Public Health Wales; NHS Wales Shared Services; DHCW, formerly called National Wales Informatics Service) to identify defined categories of patients.
▶ Identifying additional patients within the NHS (via general practice registers, outpatient lists, etc).
▶ Managing and sending out shielding advice letters (NHS Wales Shared Services; NHS Wales Delivery Unit).
▶ Sharing lists and sending out support letters to shielded people from local authorities (NHS Wales Shared Services/Delivery Unit; Unitary Authorities across Wales).
▶ Sharing and sending messages to shielded people from supermarkets for prioritised delivery slots (NHS Wales Shared Services/Delivery Unit/Supermarkets).

► Providing food parcel deliveries/pharmacy prescription pick-ups and delivery (Unitary Authorities).

We will collect these data through interviews and documentary evidence from key informants at the organisations which collaborated in delivering this intervention. We will describe the consequences of shielding in terms of healthcare resource use based on patient-level linked data extracted from SAIL (and costed using published unit costs) and COVID-19-related morbidity and mortality and summarise net monetary benefits of shielding.

### Objective 6: experiences and views of healthcare providers

We will interview 30 clinicians including general practitioners (GPs), primary care and community nurses, emergency department (ED), emergency ambulance and intensive care staff across locations in Wales. We will use vignettes developed from the interviews with the shielded population (objective 4) to understand challenges, particularly patient-reported safety concerns.[39] Questions will include views on shielding, how it was implemented and how they felt it affected people including any health risks for patients (online supplemental appendix 3). Our public contributors will codevelop these tools. Interviews will be online or by telephone, recorded and transcribed with their consent.

### Analysis
### Objective 1: rationale for UK shielding

We will analyse data using framework analysis, recommended for use in policy and health services research.[40 41] A senior qualitative researcher (AP) will lead a team of researchers and public contributors in reading and coding data for discussion and interpretation. Through this process, we will refine the logic model for the shielding intervention and understand the intentions of applying it—making explicit the hypothesised mechanisms for change, expected outcomes and risks.[32 33] This model will guide interpretation of study findings including mechanism and outcomes data and dissemination.

### Objective 2: effectiveness of shielding

We will analyse quantitative data following 'intention to treat' analysis principles. Our detailed statistical analysis plan, compliant with Swansea Trials Unit's Standard Operating Procedure,[42] will cover: descriptive summaries of study data and thematic categorisation; formal comparison of outcomes, adjusted for case mix and potential confounding factors; statistical modelling strategy underpinning comparisons, including conventions for dealing with missing data, selection of confounders; reporting of analyses. Modelling will use generalised linear and survival multilevel models for events, counts and time to events. Entry dates are based on the date identified as CEV (shielded cohort) or 20 March 2020 (matched cohort); 12-month follow-up data will be censored by death, or known date of migration from Wales.

There was considerable spatial as well as temporal variation in the (estimated) R number across Wales during our study window, with little detailed data on the accuracy of these estimates. Although difficult to justify incorporation of such estimates into formal models, we will use what is known to inform discussion of our results. We will assess if available anonymised residential identifiers allow creation of usable household/residential clusters, and, if so, whether extension to include clustering improves our models. We will consider further use of care home residence identifiers and critical care and hospital in-patient spells in defining potential explanatory covariates and factors for inclusion in models.

### Objective 3: impact of immunological status on outcomes in the shielded population

The majority of people in shielded groups had FBCs and LFTs in recent years. The all-Wales Results Reporting Service data that contain all laboratory tests on the entire population of Wales flow into the SAIL Databank and the Medical Research Council (MRC)-funded ConCOV population cohort (controlling COVID-19 through enhanced population surveillance and intervention project).[8]

These widely collected data will allow an initial broad immunological analysis of humoral immunity (using calculated globulin), cellular immunity (using lymphocyte counts) and impact due to neutropenia (neutrophil counts).

To analyse immunological data we will first plot calculated globulin from low to high in g/L increments against infection, hospitalisation, death and the same analysis for lymphocytes and neutrophils 0.1, 0.2 upwards. Using primary and secondary care data we will define groups within the shielded population, into those with none, or frequent infections to assess whether prior infection frequency relates to outcomes during the pandemic. From the dried blood sample, we will undertake COVID-19 antibody assays, which may include testing for the receptor-binding domain of the spike protein and for nucleocapsid. We will measure T-cell responses to SARS-CoV-2 using a commercially available whole-blood assay (ImmunoServ).[43] Briefly, this means 10 mL venous blood samples are collected into sodium heparin vacutainers (BD) and stimulated with a SARS-CoV-2 peptide pool containing peptides spanning the entire spike (S1 and S2) protein, nucleocapsid phosphoprotein and membrane glycoprotein for 20–24 hours at 37°C prior to a 2 min centrifugation at ×3000g. The plasma from the top of each blood sample will be harvested and analysed for interferon gamma by ELISA. This will determine levels of T-cell immunity among participants who show no antibodies to COVID-19, even though they have had either natural COVID-19 infection or vaccination to COVID-19. A positive SARS-CoV-2-specific T-cell response will be defined as >23.55 pg/mL IFNg and 50% above the negative (unstimulated) control value, as previously determined in healthy donors.[43] Tests will be run on the day of sample receipt to avoid deterioration.

## Objectives 4+6: behaviour, adherence and safety concerns of people who shielded; experiences and views of healthcare providers

We will undertake thematic analysis of interviews with people who shielded and healthcare professionals.[44] Our analysis team will include public contributors and clinical experts alongside experienced researchers. Analysis will be informed by the logic model (objective 1).[32 33] Where appropriate, anonymous excerpts will be included in reports and peer-review papers.

## Objective 5: costs of the shielding intervention against its consequences

We will estimate NHS healthcare resource use through anonymised linked data, compared between shielded and non-shielded matched cohorts. ED attendances, hospital admissions, length of stay, ICU admissions and GP contacts (if available) will be accessed within the SAIL Databank and costed using published unit costs.[45 46] Cost–consequences analysis will compare costs of shielding (including implementation cost and changes in healthcare costs) with COVID-19-related outcomes such as morbidity, mortality and health-related quality of life based on SF-12 questionnaire responses[34] from a stratified random sample of people from shielded and control cohorts. Net monetary benefit will be calculated to weigh up all costs and outcomes of the intervention.

## Study design limitations

As the shielding intervention was introduced across all the UK at one timepoint, we are able to carry out a quasi-experimental study only, with no clear historic or concurrent control group. In this circumstance, we acknowledge that any method to identify a matched comparator group is, to some extent, flawed. As entire clinical codes were allocated to the CEV (shielded) group, we intend to match as well as we can, from routine data sources, by age, sex and health service utilisation in the year prior to introduction of the shielding policy. We will then compare clinical, demographic and socioeconomic characteristics of our two groups and adjust for differences in our analysis. Without any group for comparison of outcomes it is difficult to draw any conclusions related to the benefits, harms and costs of the shielding policy. We have, therefore, selected this study design as the best available for this study.

The study was designed before widespread availability of vaccinations. We do not intend to include this variable in our data collection and analyses; so many variations in timing, vaccination delivered, number of vaccinations and boosters means this analysis is outside the scope of the current study.

Although not all those included in the shielding intervention will have received letters or complied with advice, and some people in the non-shielding cohort may have strictly self-isolated, we are following principles of analysis by intention to treat or treatment allocated[47] as this is most suited to a pragmatic evaluation context where study findings relate to how the intervention was implemented, not just how it was intended.

## Public involvement

People affected by the shielding policy have been directly involved throughout study development. Two were coapplicants on the funding proposal and are members of the Research Management Group (RMG) overseeing study implementation (LB, LD). Academic coapplicants (HS and BAE) were personally directly or indirectly affected by implementation of the shielding policy. We will recruit a Patient Advisory Panel of up to eight individuals affected by the shielding policy to supplement public input and support LD and LB. Public contributors will be involved at all stages of study delivery and dissemination. We will recruit two additional individuals to join the independent Study Steering Committee of clinical, policy, academic, methodological and public contributor experts. We will provide honoraria, briefings and other support as needed in line with best practice and report public involvement in our outputs.[48–50] We have a named lead for public involvement in the team (BAE) who brings expertise and experience to this role.

## Study management and delivery

We will implement a comprehensive strategic and operational management, delivery and oversight infrastructure: RMG (research staff, all coapplicants), bi-monthly; Patient Advisory Panel (eight public contributors; chaired by a public contributor from, and reporting to, the RMG), quarterly; independent Study Steering Committee (clinical, policy, academic, methodological and public contributor experts), half-yearly; Core Research Group, reporting to RMG, 2–4 weeks.

## ETHICS AND DISSEMINATION

We have ethical permission from the Newcastle North Tyneside 2 Research Ethics Committee (IRAS 295050) and approval under the SAIL independent IGRP project number 0911.

We will prepare a publication and engagement plan, informed by the insight and expertise of our clinical, academic, public and policy coapplicants to reach a range of audiences.

We will disseminate results directly to policy-makers through the Welsh Government COVID-19 Technical Advisory Group, and the UK government Scientific Advisory Group for Emergencies and its related subgroups.

We will publish in peer-reviewed scientific journals and present at scientific and policy conferences (for a recent example, see https://hsruk.org/conference/conference-2021/workshops/pros-and-cons-shielding-vulnerable-people-public-health-policy given to the 2021 Health Services Research UK Conference).

Our public contributors will lead production of accessible summaries of findings which we will publish online (http://www.primecentre.wales/), share with our strong

public and patient networks and promote through our social media networks.

In this first national evaluation of the effects of the UK COVID-19 shielding policy, we will contribute evidence for the role of immunity in prediction of outcome. Alongside emerging evidence from other studies undertaken through the National Core Studies Immunity programme, the proposed research will support the UK in preparation for future pandemics particularly concerning the health and safety of the most vulnerable members of society.

**Acknowledgements** This study makes use of anonymised data held in the Secure Anonymised Information Linkage (SAIL) Databank. This work uses data provided by patients and collected by the National Health Service (NHS) as part of their care and support. We would also like to acknowledge all data providers who make anonymised data available for research. We wish to acknowledge the collaborative partnership that enabled acquisition and access to the deidentified data, which led to this output. The collaboration was led by the Swansea University Health Data Research UK team under the direction of the Welsh Government Technical Advisory Cell and includes the following groups and organisations: the SAIL Databank, Administrative Data Research Wales, Digital Health and Care Wales, Public Health Wales, NHS Shared Services Partnership and the Welsh Ambulance Service Trust.

**Contributors** BAE drafted the manuscript with editorial input from all authors—AA, RB, LB, SB, AC-S, LD, AE, AJ, SJ, MRK, JL, RL, AP, BS, CAT, AW, TW, HS. The research idea was conceived by HS and developed by all authors. All authors read and approved the final manuscript.

**Funding** This work is supported by the National Core Studies Immunity (NCSi4P) Programme (award number UoB WT Ref: 1745068). This work was supported by the Con-COV team funded by the Medical Research Council (grant number: MR/V028367/1). This work was supported by Health Data Research UK, which receives its funding from HDR UK (HDR-9006) funded by the UK Medical Research Council, Engineering and Physical Sciences Research Council, Economic and Social Research Council, Department of Health and Social Care (England), Chief Scientist Office of the Scottish Government Health and Social Care Directorates, Health and Social Care Research and Development Division (Welsh Government), Public Health Agency (Northern Ireland), British Heart Foundation (BHF) and the Wellcome Trust. This work was supported by the ADR Wales programme of work (award number ES/S007393/1). The ADR Wales programme of work is aligned to the priority themes as identified in the Welsh Government's national strategy: Prosperity for All. ADR Wales brings together data science experts at Swansea University Medical School, staff from the Wales Institute of Social and Economic Research, Data and Methods (WISERD) at Cardiff University and specialist teams within the Welsh Government to develop new evidence which supports Prosperity for All by using the SAIL Databank at Swansea University, to link and analyse anonymised data. ADR Wales is part of the Economic and Social Research Council (part of UK Research and Innovation) funded ADR UK (grant ES/S007393/1). This work was supported by the Wales COVID-19 Evidence Centre, funded by Health and Care Research Wales.

**Competing interests** RL, SJ, AJ and AE are members of the Welsh Government COVID-19 Technical Advisory Group. AJ is also co-chair of the Scientific Pandemic Insights Group on Behaviours, which is a subgroup of the Scientific Advisory Group for Emergencies advising the UK government. SJ is also a member of the Welsh Government Testing Technical Advisory Group and Cardiff University COVID Strategic Advisory Board.

**Patient and public involvement** Patients and/or the public were involved in the design, or conduct, or reporting, or dissemination plans of this research. Refer to the Methods and analysis section for further details.

**Patient consent for publication** Not applicable.

**Provenance and peer review** Not commissioned; externally peer reviewed.

of the translations (including but not limited to local regulations, clinical guidelines, terminology, drug names and drug dosages), and is not responsible for any error and/or omissions arising from translation and adaptation or otherwise.

**ORCID iDs**
Bridie Angela Evans http://orcid.org/0000-0003-0293-0888
Ashley Akbari http://orcid.org/0000-0003-0814-0801
Andrew Carson-Stevens http://orcid.org/0000-0002-7580-7699
Adrian Edwards http://orcid.org/0000-0002-6228-4446
Ann John http://orcid.org/0000-0002-5657-6995
Jane Lyons http://orcid.org/0000-0002-4407-770X
Ronan Lyons http://orcid.org/0000-0001-5225-000X
Alison Porter http://orcid.org/0000-0002-3408-7007
Alan Watkins http://orcid.org/0000-0003-3804-1943
Tony Whiffen http://orcid.org/0000-0002-9329-6685

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
