## [Reviewer comments · BMJ Open]

ARTICLE DETAILS

TITLE (PROVISIONAL)	Evaluation of the Shielding Initiative in Wales (EVITE Immunity):Protocol for a quasi-experimental study
AUTHORS	Evans, Bridie; Akbari, Ashley; Bailey, Rowena; Bethell, Lesley; Bufton, Samantha; Carson-Stevens, Andrew; Dixon, Lucy; Edwards, Adrian; John, Ann; Jolles, Stephen; Kingston, Mark; Lyons, Jane; Lyons, Ronan; Porter, Alison; Sewell, Bernadette; Watkins, Alan; Whiffen, Tony; Snooks, Helen

VERSION 1 – REVIEW

REVIEWER	Colhoun, Helen The University of Edinburgh, Institute of Genetics and Cancer
REVIEW RETURNED	14-Feb-2022

GENERAL COMMENTS	Explain to the reader what you mean by a logic model What is the rationale for matching for health care utilisation – surely the point is that by definition those warranting shielding have higher utilisation – linked to their frailty – and also as we have shown hospital exposure during the pandemic was associated with subsequent risk of infection – please explain and discuss the implications of matching on the interpretability of the findings. There is not enough detail given of the time period over which this evaluation will occur or how vaccination will be considered in the analysis. The effectiveness of shielding cannot be evaluated with this design – you do not know what the risk in those who were assigned to shielding would have been had they not been assigned to the programme – a comparison with a matched cohort will tell you what relative risks were actually experienced but it will not tell you anything about shielding effectiveness – even if the relative risks remain high (which we already know they did) this does not necessarily mean shielding was ineffective. I think the best handle you could get on it would be to examine the differences in those who were and were not compliant but that will require careful adjustment for differences in prior frailty between compliers and non-compliers. Please re-word this section as it is definitely not going to evaluate effectiveness but simply the absolute and relative risks experienced. If you can identify substantial numbers who should have been assigned to the programme but through administrative error were not so assigned then once again you may be able to get some measure of effectiveness. In your analysis of this section you should explain exactly what statistical model you will use how you will define entry and exit dates and how you will handle calendar time variation in the background R number. You also need to explain how you will ascertain and deal with household size, children in household and
--

	exposure to high risk settings such as being in a care home, other long term care and hospital inpatients care. I think a protocol paper would be expected to have at least the rudimentary elements of a statistical plan included. Regarding T Cell immunity not data supporting the use of the cited assay and detailed of the required sample prep are not given and I doubt that this is feasible. Also rewording to make very clear when it is prior immunity non specific to COVID-19 as a risk factor for subsequent infection versus subsequent immunity to COVID-infection that you are trying to evaluate here would be useful. Regarding costs of shielding there is nothing in this on the costs to the shielded persons in terms of lost income child care and other care costs etc and this seems an important omission. The paper would benefit with statement on the likely limitations in the design and the certainty with which conclusion can be reached. You may find it useful to read this paper https://pubmed.ncbi.nlm.nih.gov/34158021/
--	---

REVIEWER	Brill, Simon Royal Free London NHS Foundation Trust, Respiratory Medicine
REVIEW RETURNED	21-Feb-2022

GENERAL COMMENTS	Overall this is an important, broad study and well written and described. It is observational, exploratory in some aspects and includes both qualitative and quantitative analysis. I accept therefore that there is some uncertainty about the exact statistical analysis plan and would expect this to be described in more detail in any eventual publications. I think the data generated should be very useful for future pandemic planning.
---

VERSION 1 – AUTHOR RESPONSE

Comments from Reviewer: 1 - Prof. Helen Colhoun, The University of Edinburgh

1.1 Explain to the reader what you mean by a logic model.

Our response:

We have included a short explanation of the term logic model, alongside references 31 and 32. The new text, on page 9 reads:

‘including: components of the intervention; mechanisms for change (how the intervention was expected to work); expected outcomes and impacts, including harms. The draft logic model was prepared by the study team based on published information and invite comments.’

1.2 What is the rationale for matching for health care utilisation – surely the point is that by definition those warranting shielding have higher utilisation – linked to their frailty – and also as we have shown hospital exposure during the pandemic was associated with subsequent risk of infection – please explain and discuss the implications of matching on the interpretability of the findings.

Our response:

We acknowledge that any method of identifying matched controls is somewhat flawed. We believe we have chosen the best method available to us, using routine data sources. We have included the rationale for matching by prior health service utilisation rates. This additional text, on pages 9 and 10, is as follows:

‘We know that those warranting shielding will have higher utilisation rates. The rationale for matching on health care utilisation is to identify people who should have been shielding (but were missed due to initial selection of conditions based on prioritisation for flu vaccination and/or administrative error) who

can be matched on propensity to create as similar as possible a comparator group (in the absence of randomisation) and utilise a difference in difference approach to estimate effectiveness, comparing pre- and post-health service utilisation rates.'

We have also added a new entitled 'Study design limitations' on pages 15-16, which includes the following text (on page 15) in response to this comment.

'As the shielding intervention was introduced across all the UK at one timepoint, we are able to carry out a quasi-experimental study only, with no clear historic or concurrent control group. In this circumstance, we acknowledge that any method to identify a matched comparator group is, to some extent, flawed. As entire clinical codes were allocated to the CEV (shielded) group, we intend to match as well as we can, from routine data sources, by age, sex and health service utilisation in the year prior to introduction of the shielding policy. We will then compare clinical, demographic and socio-economic characteristics of our two groups and adjust for differences in our analysis. Without any group for comparison of outcomes it is difficult to draw any conclusions related to the benefits, harms and costs of the shielding policy. We have therefore selected this study design as the best available for this study.'

1.3 There is not enough detail given of the time period over which this evaluation will occur or how vaccination will be considered in the analysis.

Our response (re: timeperiod):

We have now included the exact timeframe for recruitment and follow up. Please see the following amended text on page 10.

'- from date of their addition to the shielding list between March 23rd - December 31st 2020; and from March 23rd 2020 for the matched control groups: with follow-up of outcomes up to 1 year – '

Our response (re: vaccination):

We do not intend to include vaccination as a variable in our analysis and have included this in our new 'Study design limitation' section. Please see the following text on page 15.

'The study was designed before widespread availability of vaccinations. We do not intend to include this variable in our data collection and analyses; so many variations in timing, vaccination delivered, number of vaccinations and boosters means this analysis is outside the scope of the current study.'

1.4 The effectiveness of shielding cannot be evaluated with this design – you do not know what the risk in those who were assigned to shielding would have been had they not been assigned to the programme – a comparison with a matched cohort will tell you what relative risks were actually experienced but it will not tell you anything about shielding effectiveness – even if the relative risks remain high (which we already know they did) this does not necessarily mean shielding was ineffective. I think the best handle you could get on it would be to examine the differences in those who were and were not compliant but that will require careful adjustment for differences in prior frailty between compliers and non-compliers. Please re-word this section as it is definitely not going to evaluate effectiveness but simply the absolute and relative risks experienced.

Our response:

We believe we are using the best study design available in the circumstances of evaluation of a policy that was introduced across a whole country at the same point in time. We have added a further paragraph to our new 'Study design limitations' section, as follows, on page 16.

'Although not all those included in the shielding intervention will have received letters or complied with advice, and some people in the non-shielding cohort may have strictly self-isolated, we are following principles of analysis by Intention to Treat or Treatment Allocated [47] as this is most suited to a pragmatic evaluation context where study findings relate to how the intervention was implemented, not just how it was intended.'

1.5 If you can identify substantial numbers who should have been assigned to the programme but through administrative error were not so assigned then once again you may be able to get some measure of effectiveness.

Our response:

Thank you for this suggestion. This is what we have planned to do. We expect to find substantial numbers who were not assigned - given the way the conditions were chosen and the slightly chaotic introduction – making evaluation possible. We are not aware of another method of achieving this.

1.6 In your analysis of this section you should explain exactly what statistical model you will use how you will define entry and exit dates and how you will handle calendar time variation in the background R number. You also need to explain how you will ascertain and deal with household size, children in household and exposure to high risk settings such as being in a care home, other long term care and hospital inpatients care. I think a protocol paper would be expected to have at least the rudimentary elements of a statistical plan included.

Our response:

We have added more information to address the reviewer's helpful suggestion. The following extra text is provided on page 13.

'Modelling will use generalised linear and survival multilevel models for events, counts and time to events. Entry dates are based on the date identified as CEV (shielded cohort) or 20 March 2020 (matched cohort); 12-month follow-up data will be censored by death, or known date of migration from Wales.

There was considerable spatial as well as temporal variation in the (estimated) R number across Wales during our study window, with little detailed data on the accuracy of these estimates. Although difficult to justify incorporation of such estimates into formal models, we will use what is known to inform discussion of our results. We will assess if available anonymised residential identifiers allow creation of usable household/residential clusters, and, if so, whether extension to include clustering improves our models. We will consider further use of Care Home residence identifiers and Critical Care and Hospital in-patient spells in defining potential explanatory covariates and factors for inclusion in models.'

1.7 Regarding T Cell immunity not data supporting the use of the cited assay and detailed of the required sample prep are not given and I doubt that this is feasible. Also rewording to make very clear when it is prior immunity non specific to COVID-19 as a risk factor for subsequent infection versus subsequent immunity to COVID-infection that you are trying to evaluate here would be useful.

Our response:

We have added further information in response to the reviewer's helpful comments. On page 14 we have added the following text:

'We will measure T cell responses to SARS-CoV-2 using a commercially available whole-blood assay (ImmunoServ Ltd) [43]. Briefly, this means 10ml venous blood samples are collected into sodium heparin vacutainers (BD) and stimulated with a SARS-CoV-2 peptide pool containing peptides spanning the entire spike (S1 and S2) protein, nucleocapsid phosphoprotein and membrane glycoprotein for 20-24 hours at 37°C prior to a 2 min centrifugation at 3000 g. The plasma from the top of each blood sample will be harvested and analysed for interferon gamma by ELISA. This will determine levels of T cell immunity among participants who show no antibodies to COVID-19, even though they have had either natural COVID-19 infection or vaccination to COVID-19. A positive SARS-CoV-2-specific T cell response will be defined as >23.55pg/ml IFNg and 50% above the negative (unstimulated) control value, as previously determined in healthy donors [43].'

1.8 Regarding costs of shielding there is nothing in this on the costs to the shielded persons in terms of lost income child care and other care costs etc and this seems an important omission.

Our response:

We will include NHS costs and cost consequences only. We have clarified this now in the text. Please see page 15 under the subheading Objective 5.

1.9 The paper would benefit with statement on the likely limitations in the design and the certainty with which conclusion can be reached.

Our response:

We have now added a section on 'Study design limitations' in response to reviewers' comments. This section is on pages 15 and 16.

1.10 You may find it useful to read this paper

<https://eur03.safelinks.protection.outlook.com/?url=https%3A%2F%2Fpubmed.ncbi.nlm.nih.gov%2F34158021%2F&data=05%7C01%7CB.A.Evans%40Swansea.ac.uk%7Ce3806b1fc23847ff141a08da334ab3cb%7Cbbcab52e9fbe43d6a2f39f66c43df268%7C0%7C0%7C637878693179255580%7CUnknwn%7CTWFpbGZsb3d8eyJWljoimc4wLjAwMDAiLCJQljoiv2luMzliLCJBTiI6lk1haWwiLCJXVCI6Mn0%3D%7C3000%7C%7C&sdata=5NqjCbKXVenvEseldEVgYEW54aAqoKVGySwD8MkMw34%3D&reserved=0>

Our response:

Thank you for sending the link to this useful paper which we have read with interest. We have cited it (reference 23) on page 7 of our protocol paper and will also use it in our following outputs

Comments from Reviewer: 2 - Dr. Simon Brill, Royal Free London NHS Foundation Trust

2.1 Overall this is an important, broad study and well written and described. It is observational, exploratory in some aspects and includes both qualitative and quantitative analysis. I accept therefore that there is some uncertainty about the exact statistical analysis plan and would expect this to be described in more detail in any eventual publications. I think the data generated should be very useful for future pandemic planning.

Our response:

Thank you for these supportive comments

VERSION 2 – REVIEW

REVIEWER	Colhoun, Helen The University of Edinburgh, Institute of Genetics and Cancer
REVIEW RETURNED	13-Jun-2022

GENERAL COMMENTS	You have improved the paper and I hope you found the comments useful. I still think that you will need to be very careful in the interpreting the results of this study and be careful to describe the limits on your ability to make inference about the effectiveness of shielding - nonetheless i think some useful data will be produced by it within these constraints and I wish you well for the work
--